# The dolutegravir failure cohort: A multi-country longitudinal cohort with a randomised clinical trial of continued dolutegravir versus switch to darunavir in people with viraemia while on dolutegravir in Sub-Saharan Africa (The Ndovu Study) protocol

Loice Achieng Ombajo[1,2]*, Joseph Nkuranga[1,2], Jeremy Penner[2,3], Emily Wangui Kamau[1], Nalia Ismael[4], Patricia Munseri[5], Irene Ayakaka[6], Niklaus Labhardt[7], Dalton Wamalwa[8], Patricia Ramgi[4], Raquel Matavele Chissumba[4,9], James Wagude[10], Muhammad Bakari[5], Victor Omodi[1,2], Edwin Otieno[1,2], Lisa Abuogi[11], Rena Patel[12], Charles Opondo[13], Daniel Grint[13], Leonard King'wara[14], AnneMarie Macharia[15], Andrew Mulwa[16], Patrick Amoth[17], on behalf of the Ndovu study group¶

1 Department of Clinical Medicine and Therapeutics, University of Nairobi, Nairobi, Kenya, 2 Center for Epidemiological Modelling and Analysis, University of Nairobi, Nairobi, Kenya, 3 Department of Family Practice, University of British Columbia, Vancouver, Canada, 4 Instituto Nacional de Saúde, Marracuene, Mozambique, 5 Department of Internal Medicine, Muhimbili University of Health and Allied Sciences, Dar es Salaam, Tanzania, 6 SolidarMed – Partnerships for Health, Maseru, Lesotho, 7 Division of Clinical Epidemiology, Department of Clinical Research, University Hospital Basel, Basel, Switzerland, 8 Department of Paediatrics, University of Nairobi, Nairobi, Kenya, 9 Centro de Investigação e Desenvolvimento em Etnobotânica, Namaacha, Mozambique, 10 Siaya County Referral Hospital, Siaya, Kenya, 11 Department of Pediatrics, University of Colorado, Aurora, Colorado, United States of America, 12 Department of Medicine, Heersink School of Medicine, University of Alabama at Birmingham, Birmingham, Alabama, United States of America, 13 Faculty of Epidemiology and Population Health, London School of Hygiene and Tropical Medicine, London, United Kingdom, 14 Kenya National Public Health Institute, Nairobi, Kenya, 15 Department of Paediatrics, Kenyatta National Hospital, Nairobi, Kenya, 16 National AIDS/STI Control Program, Ministry of Health Kenya, Nairobi, Kenya, 17 Ministry of Health, Nairobi, Kenya

¶ Membership of the Termite Genome Working Group is listed in the Acknowledgments.
* loisea@uonbi.ac.ke

## Abstract

### Background

There is insufficient data to inform the management of dolutegravir failure, with the WHO and various countries adopting different approaches, underscoring the need for an evidence-based management approach.

### Methods

The Ndovu study is a large multi-country cohort, with a nested randomised controlled trial (RCT), enrolling 6,600 people living with HIV (PLWH) with viral load (VL) of ≥1000 copies/ml after at least 6 months of dolutegravir. Participants aged ≥ 1

**Data availability statement:** No datasets were generated or analysed during the current study. All relevant data from this study will be made available upon study completion.

**Funding:** This study is funded by the Gates Foundation (Investment 051835), which was involved in study design but will not be involved in the study conduct, data collection, analysis or decision to publish.

**Competing interests:** I have read the journal's policy and the authors of this manuscript have the following competing interests: Loice Achieng Ombajo.: ViiV Healthcare (research support for investigator-initiated clinical trials; participation in scientific advisory board), Gilead Sciences (research support for investigator-initiated clinical trial), participation in scientific advisory boards for GSK and MSD; Jeremy Penner.: ViiV Healthcare and Gilead Sciences (research support for investigator-initiated clinical trials). All other authors report no potential conflicts. All authors have submitted the ICMJE Form for Disclosure of Potential Conflicts of Interest.

year, including pregnant women, will be followed up for 12 months with enhanced adherence counselling (EAC) provided monthly. Viral load (VL) testing will be conducted every 3 months and drug resistance testing conducted if VL ≥ 200 copies/ml. Three hundred and sixty-two participants aged ≥15 years and 30 participants aged 3–14 years with major dolutegravir-associated drug resistant mutations (DRMs) will be enrolled into the RCT and randomised to switch to ritonavir boosted darunavir (DRV/r) or continue with dolutegravir with follow-up for 12 months. VL will be measured at 1, 3, 6 and 12 months and tenofovir levels assessed on dried blood spots at month 1 and month 6.

The primary outcome of the cohort is the proportion of participants achieving viral load <200 copies/ml by month 12 and the primary endpoint of the RCT is viral load <200copies/ml at 6 months using a modified FDA snap shot algorithm. Secondary endpoints are DRM patterns associated with non-suppression, level of adherence associated with suppression as well as participant and provider experiences of staying on DTG versus switching to DRV/r.

The RCT primary efficacy analysis will be conducted on the Intent-to-Treat Exposed (ITT-E) population and will compare the difference in the proportion of participants with viral load <200 copies/ml 6 months after randomisation between the treatment arms stratified by the randomisation stratification factors.

This study is registered at ClinicalTrials.gov, NCT06762054 (cohort) and NCT06747507 (RCT) and enrollment into the cohort started in March 2025.

## Conclusion

The Ndovu study will address critical gaps in the management of DTG failure including the emergence, determinants and implications of DTG resistance. Further, it will evaluate the optimal ART regimens to use in the setting of DTG resistance in adults and children.

## Background

The majority of people living with HIV (PLWH) on first-line antiretroviral therapy (ART) in low- and middle-income countries are on dolutegravir (DTG)-containing regimens [1]. Current World Health Organization (WHO) guidelines recommend that people on DTG-based first line ART with HIV viral load > 1,000 copies/mL should undergo enhanced adherence counselling and, if viral load remains > 1,000 copies/mL after 3 months, they should be switched to a protease inhibitor (PI)-based second line regimen [2]. However, country guidelines vary considerably; for instance, Kenya and Botswana recommend that after intensified adherence counselling, patients with persistent viraemia > 1,000 copies/ml should receive a drug resistance test (DRT) to guide switch and the choice of the optimal second-line regimen [3,4]. Mozambique and Tanzania recommend switch to 2 nucleoside reverse transcriptase inhibitors (NRTIs) and PI after failure of a DTG-containing first-line regimen, in line with the

WHO guidelines [5,6], while South Africa [7] and Lesotho [8] take a more conservative approach that recommends retaining on DTG for at least 2 years with good adherence after which management is guided by expert opinion, and Zambia recommends switch to PI with concurrent sample collection for DRT [9].

The switch to PI has disadvantages including higher cost, higher pill burden, less convenient administration (often should be taken with food), more potential drug-drug interactions, poorer tolerability and more long-term toxicities [10]. Yet, remaining on DTG with ongoing viremia could foster additional development of DRMs and jeopardise the relevance and durability of DTG for nearly all groups of PLWH.

The WHO recommendation to switch to PI-based therapy is based on the untested assumptions that people failing DTG have selected for clinically relevant integrase inhibitor drug resistance mutations (DRMs) and are therefore more likely to achieve viral suppression with a change in regimen compared to remaining on a DTG-containing regimen. Clinical trial and cohort data show that DTG-based ART rarely leads to virological failure, and that the prevalence of emergent DTG DRMs is low among those with failure [11]. Cross-sectional data show varying rates of DTG DRMs among DRTs performed; however, the frequent lack of reporting of denominators on total number of people receiving DTG or total number of people with virological failure limit the conclusions we can make about the prevalence of DTG DRMs among people with virological failure [11]. Despite low proportions of people with virological failure and emergent DTG-associated DRMs, the absolute number of people failing DTG-based regimens has important implications for ART programs given the large number of people on DTG.

Emerging data from African countries have shown varying levels of integrase drug resistance mutations. Results from a recent cross-sectional survey in Malawi in children on DTG with confirmed virological failure, found major INSTI DRM in 16.3% of 133 samples that were successfully sequenced [12], while in Mozambique, DTG resistance was found in 19.6% of 183 samples from patients with virological failure [13]. In Kenya, surveillance samples from patients with viral non-suppression show a prevalence of up to 22.6% in ART experienced patients (on DTG as second or third line regimens) and 8.3% in those failing a first line DTG-based regimen [14]. Data from Uganda, based on laboratory-based surveillance on remnant sample, showed higher DTG resistance rates in children (6.6%) than in adults (3.9%) [15]. In Lesotho, among PLHIV taking DTG-based ART for at least 18 months and having two unsuppressed VL, prevalence of major DTG DRMs was 9.4% [16]. A mathematical modelling study from South Africa, MARISA, predicted that acquired dolutegravir resistance in people failing DTG-based regiments would rise to 46.2% in 2040 with prevalence based on duration of failure while on DTG [17], pointing to the need to quickly generate the evidence to manage DTG failure.

Most people failing first-line DTG-based regimens without DTG-associated DRMs can be expected to re-suppress without a change in regimen if adherence and potential drug interactions are addressed, and therefore would not benefit from a change to PI-based regimen [18]. For PLWH who are failing DTG-based regimens and have developed DTG-associated DRMs, there is very limited direct evidence to guide their management. Data on pathways of DTG-associated DRMs and their effects on in-vitro DTG susceptibility is accumulating [19], with substitutions at eight codons currently known to contribute to reduced DTG susceptibility, and thus considered "major" DTG-associated DRMs: 66K, 92Q, 118R, 138K/A/T, 140S/A/C, 148H/R/K, 155H and 263K [20]. In a recent scoping review, Tao and colleagues found that major INSTI-associated DRMs clustered into four signature positions including R263K, G118R, N155H and Q148H with minimal overlap. The majority of viruses had just one signature mutation, predominantly R263K. Other than G118R, the other DRMs alone were not associated with high levels of reduced DTG susceptibility and studies to determine the significance of these on clinical management are needed [21].

How the genotypic resistance patterns relate to the in-vivo virological response to a DTG-containing ART regimen is uncertain, and multiple management strategies are currently being used including: increasing DTG to twice-daily dosing, switching from DTG to a PI, adding a PI to the DTG-based regimen, among others.

Evidence for switching from DTG to a PI is extrapolated from studies where participants failed a non-DTG first-line regimen and were treated with a PI-based second line regimen. Most relevant for our proposed study population is the NADIA trial which enrolled people failing an non-nucleoside reverse transcriptase inhibitor (NNRTI) plus TDF/3TC and

randomized them to second line therapy of either DTG or DRV/r, plus either TDF/3TC or AZT/3TC [22]. At week 48, DTG-based treatment was non-inferior to DRV/r (92% of participants on DRV/r achieved viral suppression <400 copies/ml compared to 90.2% on DTG). A limitation of extrapolating these results (and those of other second-line PI studies) to the population of patients failing a first line DTG-based regimen with DTG-associated DRMs is that DTG-based first-line is simpler to take and better tolerated than EFV or NVP, with a higher barrier to resistance, so people failing the DTG-based regimen may be a sub-group with more barriers to adherence.

Real-world suppression rates of patients with documented DTG-associated DRMs have been published from national program data in Malawi [23]. Among the 24 patients with DTG resistance, 18 had been on two or more regimens prior to DTG. Of 11 patients with follow-up viral load data available after implementing DRT-based regimen changes (either 2NRTI+PI/r; 2NRTI+PI/r+DTG; 2NRTI+PI/r+DTG+DTG; 2 NRTI+DTG+DTG, nine(82%) achieved a viral load<200 copies/mL. All participants who received double-dose DTG (4/4) achieved viral suppression, and five out of seven who received PI/r or PI/r+single-dose DTG achieved viral suppression. Of two patients who remained on DTG without a change in regimen despite DTG-associated DRMs, one re-suppressed.

Routine DRT to guide ART choice after DTG failure is not available in many resource limited settings and is costly. The GIVE MOVE trial, in Lesotho and Tanzania, randomised children and adolescents with recent viremia on first-line ART to the usual care arm (which consisted of a viral load-informed treatment) or to a DRT arm in which DRT and expert review informed care; and found no significant difference in the primary outcome (a composite of death, hospitalisation, new WHO stage 4 event or VL≥50 copies/ml) between the two groups [24].

To understand the efficacy of remaining on DTG-containing ART despite having virological failure, and particularly amongst those with DTG-associated DRMs, there is need to assess outcomes based on adherence levels. Viral suppression while remaining on DTG in the presence of DRMs may depend on a threshold level of adherence, and this threshold could vary with the DRM pattern. Pharmacokinetic studies have found that intracellular tenofovir-diphosphate (TFV-DP) levels in dried blood spots (DBS) correlate with cumulative TDF drug exposure in the preceding 6–8 weeks, with threshold TFV-DP levels defined for TDF adherence levels [25,26]. TFV-DP concentrations have also shown strong association with virological suppression for study participants on Tenofovir [27], as well as association with directly measured adherence on Tenofovir [28]. Among people suppressed on TDF-containing ART, TFV-DP levels in DBS also predict future viremia [29] and development of DRMs [30].

We will evaluate the optimal approach to achieving viral suppression or re-suppression in PLWH with viremia (HIV-RNA≥1,000 copies/mL) on DTG-based ART through a prospective cohort evaluation with a nested randomized control trial assessing virological suppression among people who are maintained on DTG compared to those who switch to PI (ritonavir boosted Darunavir).

## Research question

What is the most appropriate management strategy of PLWH with sustained virologic failure on DTG, with and without DTG-associated DRMs, despite enhanced adherence counselling?

## Methods

### Study design and oversight

The Ndovu study is a prospective, multi-country, observational study with a nested randomised controlled trial. The study is designed by investigators from the University of Nairobi (UON) in Kenya in collaboration with investigators from Instituto Nacional de Saúde (INS) in Mozambique, Muhimbili University of Health and Allied Sciences (MUHAS) in Tanzania, SolidarMed in Lesotho and the London School of Hygiene and Tropical Medicine (LSHTM). Data will be collected from PLWH from HIV treatment sites within Kenya, Mozambique, Tanzania and Lesotho. The study has been approved by the ethics review committees (ERCs) in each country (Kenyatta National Hospital-University of

Nairobi ERC in Kenya, Instituto Nacional de Saúde [INS] Scientific and Technical Review and the INS Institutional Review Board in Mozambique, The Ministry of Health Research and Ethics committee in Lesotho and, the National Health Research Ethics Committee in Tanzania); further regulatory approval for the clinical trial will be provided by the Pharmacy and Poisons Board in Kenya, Comité Nacional de Bioética para Saúde (Mozambique), the Tanzania Medicine and Medical Devices Authority (TMDA in Tanzania), National Health Research Ethics Committee (Lesotho) and the London School of Hygiene and Tropical Medicine Ethics Committee (UK). Any protocol amendments will be submitted for ethics and regulatory bodies review and approval prior to implementation. All participants will provide written informed consent for this and any ancillary studies (consent form provided with the protocol). The full protocol, version 1.2 dated 24 February 2025, is available at PLOS ONE supplement. A study monitor will surveil study conduct, and an independent data and safety monitoring committee will review accruing safety and efficacy data. Independent audits will be conducted by the regulatory bodies. An international steering committee of experts has been formed to guide study implementation (the roles and composition of the DSMB and steering committees are provided in the protocol).

### Ndovu cohort

**Study population.**  Participants will be enrolled from multiple care and treatment sites in each of the participating countries. Enrolment will be conducted by HIV clinicians and nurses. Eligible participants are at least 1 year old, have been receiving a DTG-containing antiretroviral regimen for at least 6 months, have a VL of ≥1000 copies/ml in the preceding 3 months and are able to provide consent (or have parental consent). Pregnant and breastfeeding women will be included, with drug resistance testing results used to guide antiretroviral therapy. Non-pregnant women of child bearing potential will be counselled and referred for routine family planning services as per the national guidelines in each participating country and a pregnancy test done at each visit. Key exclusion criteria include concomitant treatment with a PI or a NNRTI.

**Assessments.**  Viral load results from routine program data will be used to identify potential participants with VL ≥ 1000 copies/ml who will be invited for screening and those meeting the inclusion criteria enrolled ([Fig 1]). At enrolment, a serum sample will be stored for future DRT. Visits are scheduled at 3, 6, 9 and 12 months. All participants will receive monthly enhanced adherence counselling (EAC), including screening for mental health disorders, using standardised tools with additional support provided based on patient-specific adherence challenges. Monthly EAC will be delivered in person or virtually to fit participant needs. Pill counts will be documented at each visit ([Table 1]).

**Enhanced adherence counselling.**  EAC will be delivered in-person or virtually every month by trained adherence counsellors. Each EAC session will take about 45 minutes and will use standardised tools to identify potential barriers to adherence or other reasons for virological failure. Participants requiring further psychological support will be referred for care by a clinical psychologist. Every effort will be made to track participants who miss EAC session by failing to respond to phone calls or to return to clinic, including with home visits where feasible.

**Viral load monitoring.**  Samples for VL measurement will be taken at month 3, 6, 9 and 12. Participants with VL < 200 copies/ml at any time point will have a repeat VL taken after another 3 months and if the repeat VL is < 200 copies/ml then the participant will exit the study and outcomes from routinely collected program data (viral load, loss to follow-up, death) will be documented 12–24 months after enrolment. Samples with VL ≥ 200 copies/ml will undergo reflex DRT. Participants with no DTG-associated DRM will continue to receive EAC with a repeat VL 3 months later. Participants with ≥1 DTG-associated DRM will be screened for enrolment into the Ndovu clinical trial. At baseline, plasma will be stored for all participants, this will be used for baseline drug resistance testing in the event that few or no DTG-associated DRMs are identified for enrolment into the clinical trial.

**Study outcomes.**  The primary outcome is viral suppression, defined as HIV RNA < 200 copies/ml within 12 months of enrolment. Secondary outcomes include time-to-suppression, suppression among different age and viral load

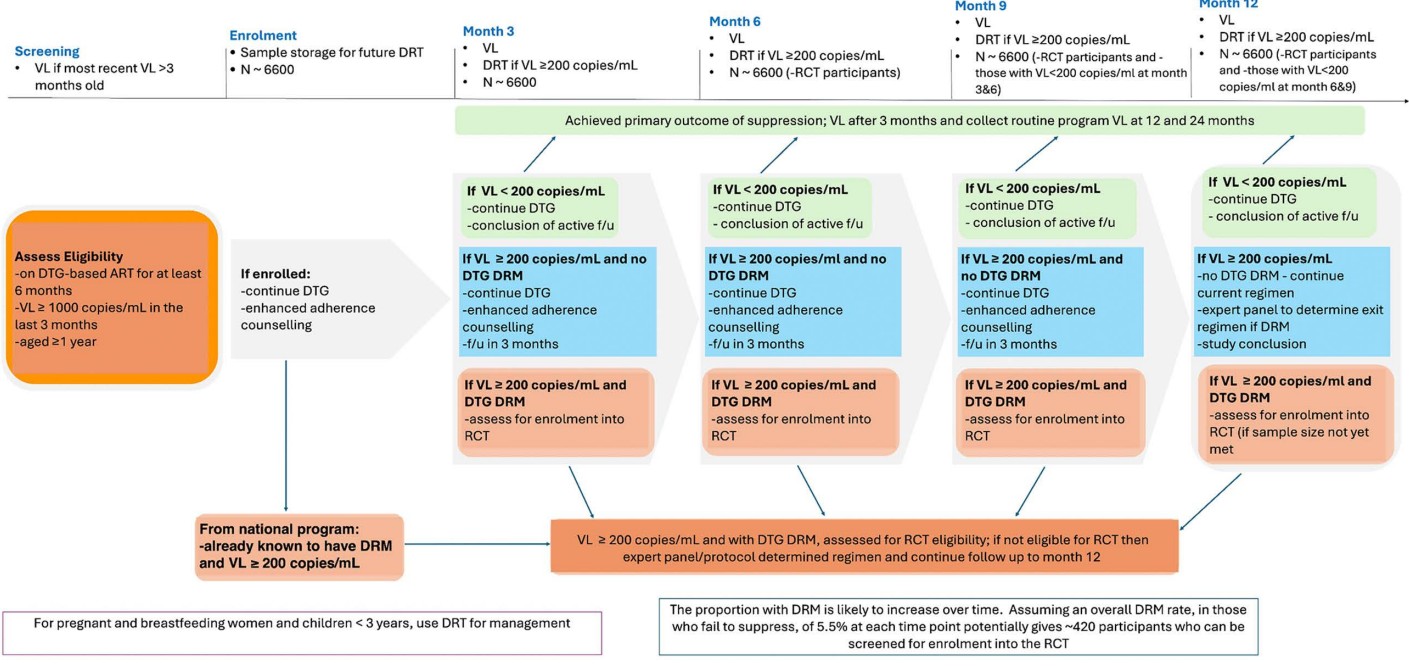

**Fig 1. Ndovu cohort study schema.**

strata, durability of viral suppression, incidence of treatment emergent DRMs and DRM patterns associated with non-suppression.

### Ndovu RCT

The Ndovu trial is a prospective, open-label, multi-center, 1:1 parallel, randomised, active-controlled, 12-month trial comparing the efficacy of DTG versus DRV/r, both in combination with tenofovir and lamivudine or emtricitabine, in participants with virological failure and at least one DTG-associated DRM while on a DTG-based regimen.

**Trial population.** We will enrol participants at 8 treatment sites in Kenya, Mozambique, Tanzania and Lesotho. Eligible participants are at least 15 years of age, are current participants in the Ndovu cohort, have a viral load of at least 200 copies/ml and at least one major DTG-associated DRM. Non-pregnant women of child-bearing potential will be counselled and referred to family planning options. We will, alongside this, enrol a small sample of children aged 3–14 years from the Ndovu cohort with VL ≥ 200 copies/ml and at least one major DTG-associated DRM. Key exclusion criteria are pregnancy and concomitant therapy with drugs that should not be used with protease inhibitors. Further details of inclusion and exclusion are described in the trial protocol which is available at PLOS supplement.

**Treatment.** Participants will be randomly assigned in a 1:1 ratio to switch to ritonavir-boosted darunavir (DRV/r) or to continue on DTG, all combined with tenofovir and lamivudine or emtricitabine. Treatment with DRV/r will be administered in the following regimens: In adults, as two tablets of co-formulated DRV/r (at doses of 400 mg and 50 mg, respectively) administered once daily plus a single co-formulated tablet of tenofovir disoproxil fumarate (TDF) and lamivudine (3TC) or emtricitabine (at doses of 300 mg and 300 mg or 200 mg respectively) administered once daily, or, a single co-formulated tablet of tenofovir alafenamide (TAF) and lamivudine or emtricitabine (at doses of 25 mg and 300 mg or 200 mg, respectively) administered once daily; in children, dosing will be based on weight (for those ≥30 kg-two tablets of co-formulated DRV/r at doses of 400 mg DRV and 50 mg ritonavir administered once daily, ≥ 15 to <30 Kg – one tablet

**Table 1. Schedule of events for the Ndovu cohort study.**

| Cohort<br>Procedure | Pre-screening | Months from enrolment | | | | |
|---|---|---|---|---|---|---|
| | | Screening/ Enrollment (Day 1)[1] | 3 | 6 | 9 | 12 |
| **Clinical and other assessments** | | | | | | |
| Written informed consent | | x | | | | |
| Inclusion/exclusion criteria | | x | | | | |
| ART history | x | x | | | | |
| Medical history (past and current) | | x | | | | |
| Concomitant medication | | x | x | x | x | x |
| Smoking, alcohol and other drug use history | | x | x | x | x | x |
| Vital signs | | x | x | x | x | x |
| Physical examination | | x | x | x | x | x |
| Height/Length[2] | | x | x | x | x | x |
| Weight | | x | x | x | x | x |
| Mid-upper arm circumference (MUAC)[3] | | x | x | x | x | x |
| Head circumference[4] | | x | x | x | x | x |
| Clinical screening for TB | | x | x | x | x | x |
| Clinical assessment for opportunistic infections | | x | x | x | x | x |
| Enhanced adherence counselling | | x | x | x | x | x |
| Dispensing of medications | | x | x | x | x | x |
| Pill counts and Proportion of Days Covered | | x | x | x | x | x |
| **Laboratory investigations** | | | | | | |
| HIV-1 RNA viral load[5] | x[6] | | x | x | x | x |
| Drug resistance test[7] | | | x | x | x | x |
| Urine pregnancy test[8] | | x | x | x | x | x |
| Stored plasma sample[9] | | x | x | x | x | x |

[1] Enrollment within 3 months of pre-screening viral load.

[2] Height for adults will be measured during the screening visit only. Length/height will be measured during each visit for participants aged 1–18 years old.

[3] Mid-upper arm circumference (MUAC) will be measured for children aged 1–5 years old.

[4] Head circumference will be measured for children under the age of 5 years.

[5] For participants achieving HIV-1 ribonucleic acid < 200 copies/mL, outcomes from routinely collected program data (mortality, loss to follow up, and viral load) will be documented 12–24 months after enrolment.

[6] Pre-screening viral load results from within 3 months prior to enrolment will be documented.

[7] Drug resistance testing will be performed for all participants with VL ≥ 200 copies/mL at months 3, 6, 9 and 12.

[8] For women of child bearing potential.

[9] Plasma will be stored for drug resistance testing if viral load is ≥ 200 copies/mL.

containing 600 mg DRV plus one tablet containing 100 mg ritonavir administered once daily, ≥ 10 to <15 Kg - one tablet of co-formulated DRV/r at doses of 400 mg DRV and 50 mg ritonavir administered once daily) and will include abacavir (ABC) in those weighing <30 kg (for those weighing between 25–30 kg- a single co-formulated tablet of ABC and 3TC at doses of 600 mg and 300 mg respectively administered once daily, and weight-band dosing using tablets with ABC co-formulated with lamivudine at doses of 120 mg and 60 mg respectively administered once daily as follows between 10–13.9 kg-two tablets, between 14–19.9 kg-two and a half tablets and between 20–24.9 kg-3 tablets). Treatment with DTG will be administered in the following regimens: as a single co-formulated tablet of DTG–lamivudine–TDF (at doses of 50 mg, 300 mg, and 300 mg, respectively) administered once daily; or as a single co-formulated tablet of DTG-lamivudine or emtricitabine-TAF (at doses of 50 mg, 300 mg or 200 mg and 25 mg, respectively) administered once daily; in children

dosing will be based on weight, with one tablet of 50 mg of DTG for those weighing 20 kg and above, and weight-band dosing using 10 mg dispersible tablets for those weighing below 20 kg (between 10–13.9 kg-two tablets and between 14–19.9 kg-two and a half tablets) and will include ABC for those weighing <30 kg (dosing as above). For children and adolescents weighing 30 kg and above, DTG will be administered as a single co-formulated tablet of DTG–lamivudine–TDF (at doses of 50 mg, 300 mg, and 300 mg, respectively) administered once daily. Switch of the NRTI is allowed for clinical indications.

Antiretrovirals will be provided through the national supply chain mechanisms of the participating countries except for DRV/r that will be purchased from the generic manufacturer, Hetero. Randomisation will be performed using R, with blocks of varying sizes and with separate sequences per the stratification categories (age group (<15 or ≥15 years old) and country) and will be uploaded on the study database for automated allocation.

**Assessments.**  Visits are scheduled at 1, 3, 6 and 12 months. Adherence will be assessed at each visit and enhanced adherence counselling provided (Fig 2). Pregnancy tests will be conducted at each visit for women of childbearing potential. Women who become pregnant will be withdrawn from the clinical trial and an expert committee will use the DRT results to define a regimen. Anthropometric measures and screening for opportunistic infections will be conducted at each visit. VL testing will be done at months 1, 3, 6, 9 and 12. Early failure will be defined as any 0.3 log rise in VL at month 1 or 3, or any new major DTG- or DRV/r-associated DRM at month 3. Any participant experiencing early failure will be withdrawn from the clinical trial and have a regimen crafted by an expert committee using the most recent DRT results; follow-up will continue to month 12. Dried blood spots will be collected and stored for measurement of TFV-DP levels at month 1 and 6, which will be batched and the test conducted at the UCT Division of Clinical Pharmacology PK Laboratory lab at the University of Cape Town, South Africa. DRT will be conducted for all samples with VL ≥ 200 copies/ml (Table 2). Clinical evaluation for adverse events will be conducted at every visit. Clinical trial insurance is provided for any harm arising from the trial.

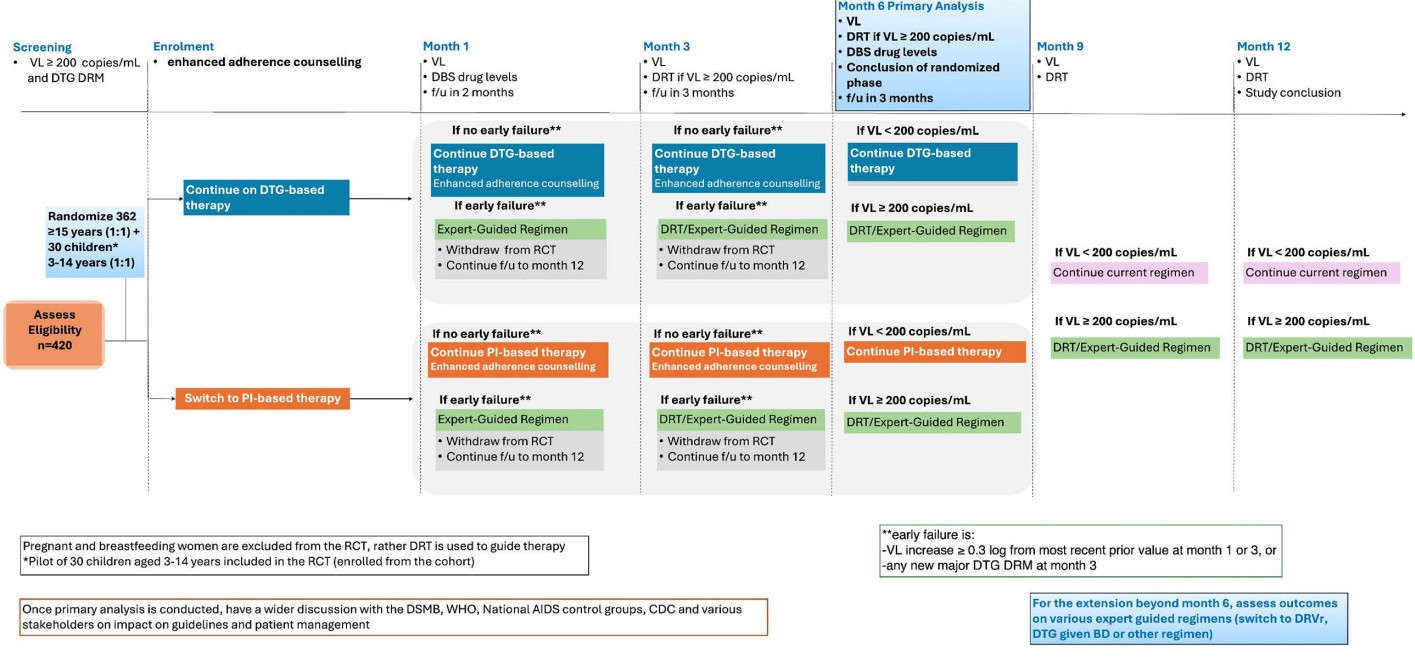

**Fig 2.  Ndovu RCT schema.**

**Table 2. Schedule of events for the Ndovu RCT.**

| RCT Procedure | Months from enrolment | | | | | |
|---|---|---|---|---|---|---|
| | Screening/ Enrollment (Day 1) | 1 | 3 | 6 | 9 | 12 |
| **Clinical and other assessments** | | | | | | |
| Written informed consent | x | | | | | |
| Inclusion/exclusion criteria | x | | | | | |
| Randomization | x | | | | | |
| ART history | x | | | | | |
| Medical history (past and current) | x | | | | | |
| Concomitant medication | x | x | x | x | x | x |
| Smoking, alcohol and other drug use history | x | x | x | x | x | x |
| Vital signs | x | x | x | x | x | x |
| Physical examination | x | x | x | x | x | x |
| Height[1] | x | x | x | x | x | x |
| Weight | x | x | x | x | x | x |
| Mid-upper arm circumference (MUAC) and Head circumference[2] | x | x | x | x | x | x |
| Clinical screening for TB | x | x | x | x | x | x |
| Clinical assessment for opportunistic infections | x | x | x | x | x | x |
| Adverse events | | x | x | x | x | x |
| Serious adverse events | | x | x | x | x | x |
| Enhanced adherence counselling | x | x | x | x | x | x |
| Dispensing of medications | x | x | x | x | x | x |
| Collection of unused medications | x | x | x | x | x | x |
| Pill counts and PDC | x | x | x | x | x | x |
| Patient satisfaction using the HIV treatment satisfaction questionnaire (HIVTSQ) | HIVTSQ status version (HIVTSQs) | | | HIVTSQ change version (HIVTSQc) | | |
| **Laboratory investigations** | | | | | | |
| HIV-1 RNA viral load[3] | x | x | x | x | x | x |
| Cr | x | | x | | | |
| AST | x | | x | | | |
| ALT | x | | x | | | |
| CD4 | x | | | x | | x |
| CBC | x | | x | | | |
| DBS for TFV-DP levels | | x | | x | | |
| Drug resistance test[4] | | | x | x | x | x |
| Urine pregnancy test[5] | x | x | x | x | x | x |

[1] Height will be measured during each visit for participants aged 1–18 years old; for participants aged more than 18 years, it will be measured once during the screening/enrollment visit.

[2] Mid-upper arm circumference (MUAC) and Head Circumference will be documented for children aged 3–5 years old.

[3] If the most recent VL from the cohort study was collected > 3 months, then do a repeat viral load during the screening/enrollment visit.

[4] Genotypic drug resistance testing will be performed for all participants with viral load ≥ 200 copies/ml.

[5] For women of child bearing potential.

**End points.** The primary endpoint of the trial is viral load <200 copies/ml 6-months after randomisation. The main secondary outcomes are VL<200 copies/ml at 12 months, VL<50 copies/ml at 6 and 12 months, treatment-emergent DRMs, TFV-DP levels in blood associated with suppression, DRM patterns associated with non-suppression and, safety.

**Data management.** Data will be collected electronically on REDCap by the study clinicians, nurses and adherence counsellors. Data quality checks conducted daily by a dedicated data management team. The data management plan is provided with the study protocol. De-identified data will be stored in password protected databases at the University of Nairobi. The Data Safety and Monitoring Board will review clinical trial data at three timepoints during the study and ad hoc as may be necessary.

## Statistical methods (Cohort and RCT)

The target sample size for the cohort study is 6,600 participants. Assuming 70% of cohort participants achieve the primary outcome of HIV-1 RNA<200 copies/mL within 12 months from enrolment, this sample size will be sufficient to estimate the primary outcome with a 1.3% precision of the 95% confidence interval. Participants who are found to have DTG-associated DRMs during cohort follow-up will be assessed for eligibility into the RCT. We have assumed that 5.5% of the cohort participants, i.e., about 363 participants will develop DTG-associated DRMs and be eligible for enrolment into the RCT, Further assuming 55% of participants will achieve VL<200 copies/mL at 6 months from randomisation on DTG, and 70% on DRV/r, then by enrolling 181 participants in each arm (362 total) the trial will have 80% power to detect this 15% absolute difference in proportion of participants achieving VL<200 copies/mL at the 5% level of significance while allowing for 10% attrition of the study population prior to the 6-month primary endpoint. In addition, we will enrol 30 children aged 3–14 years as a paediatric pilot bringing the total RCT sample size to 392 participants.

We will describe the characteristics of the cohort using counts and proportions for categorical variables and means and standard deviations or medians and ranges for continuous variables.

Overall viral suppression in the cohort will be estimated at 3, 6 and 12 months, using a generalized linear regression model of the binomial family with identify link function. We will also use a generalised linear regression model to estimate differences in the proportions with viral suppression, development of DTG-associated DRMs, development of opportunistic infections and effect of choice of NRTI on suppression. Logistic regression models will be used to assess predictors of achieving suppression, development of DTG-associated DRMs and development of opportunistic infections. Covariates to be included in the model are age (age categories for child, adolescent or adult), sex, duration on DTG, NRTI backbone, previous high viral loads and viral load at time of switch to DTG. Cox regression will be used to compare time from enrolment to development of DTG-associated DRMs in different age and viral load strata. The proportional hazards assumption will be evaluated using graphical methods (Schoenfeld residual plots) and a goodness-of-fit test (Schoenfeld Residual Tests). Baseline imbalance between study arms will be assessed, and if present, these variables will be included as covariates in the Cox regression model. Adherence levels associated with suppression and development of DTG-associated DRMs will be assessed descriptively using counts and proportions.

For the RCT, the primary efficacy analysis will be conducted on the Intent-to-Treat Exposed (ITT-E) study population, which is defined as all participants who receive at least one dose of study drug. Participants will be assessed according to the treatment regimen to which they were randomised. We will also analyse the primary endpoints using the per-protocol (PP) study population, which will exclude participants who switch treatments outside of the protocol recommendations during the course of the study. The difference in the proportion of participants with viral suppression 6 months after randomisation will be compared between the treatment arms adjusted for the study site as a fixed effect. The point estimate for the difference between the treatment strategies will provide the best available data on the comparative efficacy of the two strategies in this study population. Missing viral load data will be treated as either treatment failure or by exclusion as per FDA snapshot analysis. In addition, sensitivity analysis will include treating missing viral load data as missing at random, with multiple imputation performed using chained equations method. Baseline covariate imbalance will be assessed using

descriptive statistics (means, percentages) alongside tests for significance difference between arms (t-test, Wilcoxon). Where there is imbalance, in covariates, we will use unadjusted and adjusted regression models to ensure that confidence intervals are appropriately quantified. A protocol defined interim analysis will take place once 242 participants (two-thirds of the study population) have data available on viral suppression at 3 months, defined by HIV VL<200 copies/mL. In order for the DSMB to recommend stopping the RCT due to established efficacy, the protocol defined interim analysis must demonstrate superiority of PI-based therapy at a substantial superiority margin of 15%, beyond reasonable doubt.

Statistical analysis will be performed using R version 4.5.1.

Study results will be presented at scientific conferences and published in peer-reviewed journals. All authors will fulfil ICMJE criteria. Datasets will be published on Harvard Dataverse.

Enrolment into the Ndovu study started on 03/03/2025 with 23% of the cohort population enrolled as of 23/07/2025 and is estimated to end by 30/12/2025. Enrolment into the RCT will start on 01/09/2025 and is estimated to end on 30/06/2026. Initial results from the cohort are expected on 31/03/2026 and from the RCT on 28/02/2027.

## Discussion

The Ndovu study seeks to provide critical evidence for the management of DTG failure particularly in regard to: whether and when viremia with DTG resistance requires regimen switch; whether and when, and, in which populations drug resistance testing is required; and, whether switch to a protease inhibitor is the most appropriate option after failure on DTG.

Despite the WHO high priority recommendation on monitoring of DTG resistance through surveys, most countries are yet to take up this recommendation with only 10 countries reporting survey results [15]. Capacity for DRT is limited in most countries due to high costs and challenges with supplies and equipment maintenance [31] and is often not available to guide clinical decision making. With these limitations, it is important to identify the right population for routine DRT after virological failure. Whereas resistance in previously INSTI-naive populations has been lower than 3% as documented in multiple clinical trials [32], this may be different in those with previous ART exposure. A recent study from Mozambique in a highly select population of patients on a non-first line regimen with high VL despite multiple adherence interventions found high rates of resistance to DTG (51/57 89.5%) with major DTG-associated DRMs in 80% (46/57), with a significant number having the G118R mutation which confers the highest risk of resistance. This potentially points to a population that would benefit from DRT to inform regimen change [33].

Whereas in high resource limited settings, drug resistance testing to inform individual care [34] is recommended, this approach is not feasible in most resource limited settings and with recent cuts in global HIV funding [35], prioritisation of DRT is even more uncertain. In this context of increasing resource constraints, the information from the Ndovu study will be critically important to guide prioritisation of resources for managing DTG failure.

The strength of this study is the large sample size from multiple countries with inclusion of children and pregnant and breastfeeding women which increases its generalisability to diverse populations. We run the risk of high viral re-suppression rates in the cohort which would jeopardise enrolment into the clinical trial. Stored samples at baseline will allow us to do drug resistance tests and support subsequent analysis.

## Supporting information

**S1 File. 1a fillable-SPIRIT-outcomes-2022-checklist-with-SPIRIT-2013.**
(PDF)

**S2 File. 1b SPIRIT_fillable-checklist-15-Aug-2013.**
(PDF)

**S3 File. 1c fillable-SPIRIT-outcomes-2022-checklist.**
(PDF)

**S4 File. 2 Ndovu cohort study protocol v1.2 CLEAN.**
(DOCX)

**S5 File. 2 Ndovu RCT protocol v1.2 CLEAN.**
(DOCX)

**S6 File. 7a appendix 11 Ndovu steering committee charter 17032025.**
(PDF)

**S7 File. 7b appendix 10 DSMB charter.**
(PDF)

## Acknowledgments

Study Group team: Lazarus Momanyi, Carolyne Mwangi, Prosper Njau, Deonilde Sarmento, Tapiwa Tarumbiswa, Simon Wahome, Rukia Aksam, Aabid Ahmed, Nashina Admani, Florentius Ndinya, Elizabeth Abong'o, Dorcas Khasowa, Ricky Echesa, Antony Kiplagat, Eunice Kinywa, Muchui Christopher Ntika, Bob Awino, Stephen Oduor, Zaituni Ahmed, George Kissinger, Martha Chelangat, Wambui Karuoya, Dancan Wamati, Zian Muikamba, Andrew Njoroge, Christine Ngacha, Dorothy Mwagae, Jacquelyn Nyange, Wanjiku Ndegwa, Esther Kaloki, Amos Mulama, Robert Kisia, Teresa Kabue, Clare Iminyi, Sheila Abonyo, Christine Murugi, Simon Kamau, Ann Lilly Mbau, Dan Khaemba, Joshua Njenga, Doreen Kwamboka, Michael Odondo, Christine Ombogo, Rose Nyakoni, Frank Msafiri, Jamila Didi, Joanna Sturgess, Anne Kariithi, Antony Muchiri, Kassia Pereira, Blaise Lukau, Anna Klicpera, Laurena Urda, Cacildo Magul, Odete Bule, Lucia Manuel, Joao Manuel, Joao Alface Study group lead author: Loice Achieng Ombajo (loisea@uonbi.ac.ke) Study group affiliations:

 LM, CM - Kenya national HIV program
 PN - Tanzania national HIV program
 DS - Mozambique national HIV program
 TT - Lesotho national HIV program
 S.W - Kenyatta National Hospital
 R.A, F.N, E.A - Jaramogi Odinga Odinga Teaching and Referral Hospital
 N.A - Bomu Hospital
 A.K. C.N - Nairobi County
 E.K - Kisumu County
 B.A, S.O - Siaya County
 Z.A, G.K - Mombasa County
 D.K, R. E, M.C, W.K, D.W, Z.M, A.N, C.N, D.M, J.N, W.N, E.K, A.M, R.K, T.K, C.I, S.A, C.M, S.K, AL.M, D.K, J.N, D.K, M.O, C.O, A.K, A.M, R.M - University of Nairobi
 F.M, J.D - Muhimbili University of Health and Allied Sciences
 K.P, C.M, O.B, L.M, J.M, J.A - Instituto Nacional de Saúde, Marracuene
 B.L, A.K. L.U - Solidar Med.

## Author contributions

**Conceptualization:** Loice Achieng Ombajo, Joseph Nkuranga, Jeremy Penner, Emily Wangui Kamau.

**Data curation:** Emily Wangui Kamau, Nalia Ismael, Patricia Munseri, Edwin Otieno.

**Formal analysis:** Victor Omodi, Edwin Otieno.

**Funding acquisition:** Loice Achieng Ombajo.

**Investigation:** Joseph Nkuranga, Nalia Ismael, Patricia Munseri, Irene Ayakaka.

**Methodology:** Loice Achieng Ombajo, Joseph Nkuranga, Jeremy Penner, Emily Wangui Kamau, Nalia Ismael, Patricia Munseri, Irene Ayakaka, Niklaus Labhardt, Dalton Wamalwa, Patricia Ramgi, Raquel Matavele Chissumba, James Wagude, Muhammad Bakari, Victor Omodi, Lisa Abuogi, Rena Patel, Charles Opondo, Daniel Grint, Leonard King'wara, AnneMarie Macharia.

**Project administration:** Loice Achieng Ombajo, Joseph Nkuranga.

**Resources:** Loice Achieng Ombajo, Andrew Mulwa, Patrick Amoth.

**Supervision:** Loice Achieng Ombajo, Joseph Nkuranga, Emily Wangui Kamau, Nalia Ismael, Patricia Munseri, Irene Ayakaka.

**Writing – original draft:** Loice Achieng Ombajo.

**Writing – review & editing:** Joseph Nkuranga, Jeremy Penner, Emily Wangui Kamau, Nalia Ismael, Patricia Munseri, Irene Ayakaka, Niklaus Labhardt, Dalton Wamalwa, Patricia Ramgi, Raquel Matavele Chissumba, James Wagude, Muhammad Bakari, Victor Omodi, Edwin Otieno, Lisa Abuogi, Rena Patel, Charles Opondo, Daniel Grint, Leonard King'wara, AnneMarie Macharia, Andrew Mulwa, Patrick Amoth.

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
