## [Decision Letter · Decision Letter 0]

27 Oct 2025

Dear Dr. Ombajo,

Thank you for submitting your manuscript to PLOS ONE. After careful consideration, we feel that it has merit but does not fully meet PLOS ONE’s publication criteria as it currently stands. Therefore, we invite you to submit a revised version of the manuscript that addresses the points raised during the review process.

We look forward to receiving your revised manuscript.

Kind regards,

Sarah Nanzigu, Ph.D.,MSc.,MBchB

Academic Editor

PLOS ONE

“This work is funded by the Gates Foundation (investment 051835)”

“This study is funded by the Gates Foundation (Investment 051835) to LAO and supported by the 423 Ministries of Health of Kenya, Tanzania, Mozambique and Lesotho”

“This work is funded by the Gates Foundation (investment 051835)”

5. One of the noted authors is a group or consortium “Ndovu study group” In addition to naming the author group, please list the individual authors and affiliations within this group in the acknowledgments section of your manuscript. Please also indicate clearly a lead author for this group along with a contact email address.

Additional Editor Comments:

I am aware that a full prior description of the modelling process is often not possible at the proposal stage. However, I encourage the authors to provide little more details on the proposed analytical plan following recommendations for the first reviewer.

Reviewer's Responses to Questions

**Comments to the Author**

1. Does the manuscript provide a valid rationale for the proposed study, with clearly identified and justified research questions?

Reviewer #1: Yes

Reviewer #2: Yes

2. Is the protocol technically sound and planned in a manner that will lead to a meaningful outcome and allow testing the stated hypotheses?

Reviewer #1: Yes

Reviewer #2: Yes

3. Is the methodology feasible and described in sufficient detail to allow the work to be replicable?

Reviewer #1: Yes

Reviewer #2: Yes

4. Have the authors described where all data underlying the findings will be made available when the study is complete?

Reviewer #1: Yes

Reviewer #2: Yes

5. Is the manuscript presented in an intelligible fashion and written in standard English?

Reviewer #1: Yes

Reviewer #2: Yes

You may also provide optional suggestions and comments to authors that they might find helpful in planning their study.

Reviewer #1: As the statistical reviewer I will focus on methods and reporting. the paper is well written with enough detail on methods, which are broadly appropriate.

Major

1) Clarity is needed about the covariates in the models, what they will be or how the authors will decide what will be included.

2) all the models need to account for the clustered nature of the data through random effects: random intercept for logistic/linear models; shared frailty for Cox models.

3) clarity is needed about how the authors will deal with missing data - will multiple imputations be used and when?

4) I expected more on randomisation and blinding. All i could find was "Randomisation will be 1:1, and will employ a pre-programmed computer-generated sequence." which is not explaining this clearly, unless I missed more information in the main paper elsewhere. In addition further below the authors say "The difference in the proportion of participants with viral suppression months after randomisation will be compared between the treatment arms adjusted for the randomisation stratification factors" - what are these randomisation stratification factors. how are they incorporated into the computer generated sequence?

5) there is no information on the analytical approaches for the RCT e.g. t-test, regression model etc. I would expect to see something like "balance will be assessed and we will consider whether adjustment for covariates is needed, in which case a regression model will be used - or a simple regression if no adjustment is deemed necessary". I'd still use a regression framework (unlike say a t-test) so effect sizes are quantified and analyses are moved away from p-values.

Minor

1) describe the time to event model as "Cox regression". explain how the proportional hazards assumption will be evaluated and what covariates will you use in the model (or how you will decide what covariates to incllude) - linked to major point 1.

2) what software will be used.

Reviewer #2: The study protocol seems sound and scientifically rigorous. I don't have any further comments. I recommend it for publication.

**Do you want your identity to be public for this peer review?** For information about this choice, including consent withdrawal, please see our Privacy Policy

Reviewer #1: No

Reviewer #2: No

---

## [Author Response · Author response to Decision Letter 1]

5 Dec 2025

03-Dec-2025

Ref: PONE-D-25-41434

To: Sarah Nanzigu, PhD, MSc, MBChB

Academic Editor,

PLOS ONE

Dear Dr. Nanzigu,

Thank you for the invitation to submit a revised version of the manuscript for The dolutegravir failure cohort: a multi-country longitudinal cohort with a randomised clinical trial of continued dolutegravir versus switch to darunavir in people with viraemia while on dolutegravir in Sub-Saharan Africa (The Ndovu Study) protocol. We appreciate the edits and comments and have addressed all these, as outlined below and included in tracked changes and clean versions of the revised manuscript.

Editor’s comments

Response:

The manuscript and all accompanying files were formatted according to PLOS ONE’s style requirement

“This work is funded by the Gates Foundation (investment 051835)”

Response:

The statement on funder role was already included in the manuscript and will now be included in the cover letter. The statement reads:

This study is funded by the Gates Foundation (Investment 051835), which was involved in study design but will not be involved in the study conduct, data collection, analysis or decision to publish.

Response:

The grant number provided in the funding information is correct.

This study is funded by the Gates Foundation (Investment 051835) to LAO

“This study is funded by the Gates Foundation (Investment 051835) to LAO and supported by the Ministries of Health of Kenya, Tanzania, Mozambique and Lesotho”

“This work is funded by the Gates Foundation (investment 051835)”

Response:

We will leave the funding statement as is. The Ministries of Health of the 4 countries are providing support through provision of antiretroviral agents through the national HIV programs, but not directly funding the study

5. One of the noted authors is a group or consortium “Ndovu study group” In addition to naming the author group, please list the individual authors and affiliations within this group in the acknowledgments section of your manuscript. Please also indicate clearly a lead author for this group along with a contact email address.

Response:

The individual authors are already listed in the acknowledgements section and we have now added the affiliations. The manuscript corresponding author is also the lead author of the group and this has now been indicated.

We are not sure how to indicate the affiliations as most of the study group members are affiliated to the institutions of the main authors and the study group list is very long. We are happy to correct this if necessary.

Response:

The ethics statement is already included, we have now included the full name of the IRB committees that approved the protocol in each of the countries. The manuscript already incudes a statement on consent that reads:

All participants will provide written informed consent for this and any ancillary studies (consent form provided with the protocol).

Response:

There is no reviewer recommendation to cite any previously published works

Response:

We have reviewed the reference list and confirm that this is complete and correct.

Additional Editor Comments:I am aware that a full prior description of the modelling process is often not possible at the proposal stage. However, I encourage the authors to provide little more details on the proposed analytical plan following recommendations for the first reviewer.

Response:

We have provided responses to the first reviewer’s comments

Reviewer 1 comments

A statistical analysis plan will be written to accompany the study protocol. The analysis plan will include detailed descriptions of all the elements raised by the reviewer.

Major1) Clarity is needed about the covariates in the models, what they will be or how the authors will decide what will be included.

Response:

The covariates in the modes include: age (age categories for child, adolescent or adult), sex, duration on DTG, NRTI backbone, previous viral load, viral load at time of switch to DTG

As an individually randomised trial, the primary analysis will be adjusted for the randomisation stratification factors: study site (study country) and age category (<15 vs >15 years old. The manuscript mentions we will adjust for the stratification factors, we have added that this specifically relates to study site.

2) all the models need to account for the clustered nature of the data through random effects: random intercept for logistic/linear models; shared frailty for Cox models.

Response:

The trial is individually randomised, not cluster randomised. Therefore, random effects are not required to account for clustering in the data. The randomisation is stratified by study site and the analysis will adjust for site as a fixed effect, as detailed above.

3) clarity is needed about how the authors will deal with missing data - will multiple imputations be used and when?

Response:

Missing viral load data will be treated as either treatment failure or by exclusion as per FDA snapshot analysis. In addition, missing viral load will be treated as Missing At Random and multiple imputation performed using chained equations (MICE) method for further sensitivity analysis.

Full details of how we will handle other data will be included in the statistical analysis plan. Every attempt will be made to minimise missing data through regular follow-up visits.

4) I expected more on randomisation and blinding. All I could find was "Randomisation will be 1:1, and will employ a pre-programmed computer-generated sequence." which is not explaining this clearly, unless I missed more information in the main paper elsewhere. In addition further below the authors say "The difference in the proportion of participants with viral suppression months after randomisation will be compared between the treatment arms adjusted for the randomisation stratification factors" - what are these randomisation stratification factors. how are they incorporated into the computer generated sequence?

Response:

Randomization is performed using R using blocks of varying sizes. The stratification factors are participating countries (Kenya, Mozambique, Tanzania and Lesotho) and the participant’s age category (<15 and ≥15). This resulted in separate sequences per age group and country that were uploaded on the study database (REDCap) for automated (and blind) allocation. This has been included in the text.

This is an open-label study

5) there is no information on the analytical approaches for the RCT e.g. t-test, regression model etc. I would expect to see something like "balance will be assessed and we will consider whether adjustment for covariates is needed, in which case a regression model will be used - or a simple regression if no adjustment is deemed necessary". I'd still use a regression framework (unlike say a t-test) so effect sizes are quantified and analyses are moved away from p-values.

Response:

Baseline covariate imbalance will be assessed using descriptive statistics (means, percentages) alongside tests like t-tests/Wilcoxon for significance with the priority being clinical relevance over just the p-values. Where there is imbalance, we will use unadjusted and adjusted regression models to ensure that confidence intervals are appropriately quantified, and emphasis is placed on estimation rather than the p-values.

Minor1) describe the time to event model as "Cox regression". explain how the proportional hazards assumption will be evaluated and what covariates will you use in the model (or how you will decide what covariates to incllude) - linked to major point 1.

Response:

The proportional hazards assumption will be evaluated using graphical methods (Schoenfeld residual plots) and a goodness-of-fit test (Schoenfeld Residual Tests). Baseline imbalance between study arms will be assessed - if present, these variables will be included as covariates in the Cox regression model. However, clinical relevance will be prioritised over just the p-values from the imbalance assessment.

2) what software will be used.

Response:

Statistical analysis will be made using R version 4.5.1.

Reviewer #2: The study protocol seems sound and scientifically rigorous. I don't have any further comments. I recommend it for publication.

Thank you

We appreciate your consideration of the revised manuscript.

Yours Sincerely,

Dr. Loice Achieng Ombajo,

Principal Investigator,

Senior Lecturer,

Department of Clinical Medicine and Therapeutics,

University of Nairobi.

---

## [Decision Letter · Decision Letter 1]

23 Dec 2025

Dear Dr. Loice Achieng Ombajo,

Thank you for submitting your revised manuscript to PLOS ONE. After careful consideration, we feel that it has merit but does not fully meet PLOS ONE’s publication criteria as it currently stands. Therefore, we invite you to submit a revised version of the manuscript that addresses the points raised during the review process.

We look forward to receiving your revised manuscript.

Kind regards,

Sarah Nanzigu, Ph.D.,MSc.,MBchB

Academic Editor

PLOS One

Journal Requirements:

Additional Editor Comments:

Following editorial discussions, we kindly request you to revise your manuscript and incorporate responses to queries raised by reviewer 1.

Reviewers' comments:

Reviewer's Responses to Questions

**Comments to the Author**

1. Does the manuscript provide a valid rationale for the proposed study, with clearly identified and justified research questions?

Reviewer #1: Yes

2. Is the protocol technically sound and planned in a manner that will lead to a meaningful outcome and allow testing the stated hypotheses?

Reviewer #1: Yes

3. Is the methodology feasible and described in sufficient detail to allow the work to be replicable?

Reviewer #1: Yes

4. Have the authors described where all data underlying the findings will be made available when the study is complete?

Reviewer #1: Yes

5. Is the manuscript presented in an intelligible fashion and written in standard English?

Reviewer #1: Yes

You may also provide optional suggestions and comments to authors that they might find helpful in planning their study.

Reviewer #1: It is irrelevant if the randomisation is at the cluster level. adjusting for site clustering in the analyses addressed possible effect heterogeneity of the intervention, which is often the case.

Also the responses to me (the rest of them) were satisfying but the paper was not changed to reflect that: e.g. missing data approach, analytical approach

go through each comment i made previously and explain what was done in the paper to address it as well, don't just note your argument to me - readers may have the same questions.

**Do you want your identity to be public for this peer review?** For information about this choice, including consent withdrawal, please see our Privacy Policy

Reviewer #1: No

---

## [Author Response · Author response to Decision Letter 2]

18 Feb 2026

16 January 2026

Ref: PONE-D-25-41434R1

To: Dr. Sarah Nanzigu,

Academic Editor,

PLOS ONE

Dear Dr. Nanzigu

Thank you for the additional review of our revised manuscript “The dolutegravir failure cohort: a multi-country longitudinal cohort with a randomised clinical trial of continued dolutegravir versus switch to darunavir in people with viraemia while on dolutegravir in Sub-Saharan Africa (The Ndovu Study) protocol”.

We have addressed all comments, as outlined below and included in tracked and clean versions of the revised manuscript as appropriate.

Additional Editor Comments:

Following editorial discussions, we kindly request you to revise your manuscript and incorporate responses to queries raised by reviewer 1.

Response: We have revised the manuscript to incorporate responses to the queries raised by reviewer 1, as specified below.

Reviewer #1 Comments:

It is irrelevant if the randomisation is at the cluster level. Adjusting for site clustering in the analyses addressed possible effect heterogeneity of the intervention, which is often the case.

Response: Thank you for pointing this out. In our prior submission, the methods section of the manuscript was revised to specify that the analysis will be adjusted for the study site. We have now revised the manuscript further for additional clarity, so the sentence now reads as:

The difference in the proportion of participants with viral suppression 6 months after randomisation will be compared between the treatment arms adjusted for the study site as a fixed effect.

Also the responses to me (the rest of them) were satisfying but the paper was not changed to reflect that: e.g. missing data approach, analytical approach. Go through each comment I made previously and explain what was done in the paper to address it as well, don't just note your argument to me - readers may have the same questions.

Response: We have reviewed the comments from the first review again and have specified where the manuscript was revised to incorporate clarifications, including new revisions to the updated manuscript, point-by-point, below.

Major

Clarity is needed about the covariates in the models, what they will be or how the authors will decide what will be included.

Manuscript revision: In our prior submission, the manuscript was revised to specify the covariates that will be included in the models. The following sentence was added to the methods section of the manuscript in the prior submission:

Covariates to be included in the model are age (age categories for child, adolescent or adult), sex, duration on DTG, NRTI backbone, previous high viral loads and viral load at time of switch to DTG.

All the models need to account for the clustered nature of the data through random effects: random intercept for logistic/linear models; shared frailty for Cox models

Manuscript revision: In our prior submission, the methods section of the manuscript was revised to specify that the analysis will be adjusted for the study site. We have now revised the manuscript further for additional clarity, so the sentence now reads as:

The difference in the proportion of participants with viral suppression 6 months after randomisation will be compared between the treatment arms adjusted for the study site as a fixed effect.

Clarity is needed about how the authors will deal with missing data - will multiple imputations be used and when?

Manuscript revision: We had provided a response to this review comment in our prior submission but had not revised the manuscript to include a clarification on how missing data will be handled. We have now revised the manuscript to include the following clarification within the methods section:

Missing viral load data will be treated as either treatment failure or by exclusion as per FDA snapshot analysis. In addition, sensitivity analysis will include treating missing viral load data as missing at random, with multiple imputation performed using chained equations method.

I expected more on randomisation and blinding. All I could find was "Randomisation will be 1:1, and will employ a pre-programmed computer-generated sequence." which is not explaining this clearly, unless I missed more information in the main paper elsewhere. In addition further below the authors say "The difference in the proportion of participants with viral suppression months after randomisation will be compared between the treatment arms adjusted for the randomisation stratification factors" - what are these randomisation stratification factors. how are they incorporated into the computer generated sequence?

Manuscript revision: In our prior submission, the manuscript was revised to provide further details on randomisation, the randomisation stratification factors and how the stratification factors are incorporated into the computer-generated sequence. We have now revised the manuscript further for additional clarity, so the sentence now reads as:

Randomisation will be performed using R, with blocks of varying sizes and with separate sequences per the stratification categories (age group (<15 or ≥15 years old) and country) and will be uploaded on the study database for automated allocation.

There is no information on the analytical approaches for the RCT e.g. t-test, regression model etc. I would expect to see something like "balance will be assessed and we will consider whether adjustment for covariates is needed, in which case a regression model will be used - or a simple regression if no adjustment is deemed necessary". I'd still use a regression framework (unlike say a t-test) so effect sizes are quantified and analyses are moved away from p-values.

Manuscript revision: We had provided a response to this review comment in our prior submission but had not revised the manuscript to include a clarification on the analytical approach to the RCT. We have now revised the manuscript to include the following clarification within the methods section:

Baseline covariate imbalance will be assessed using descriptive statistics (means, percentages) alongside tests for significance difference between arms (t-test, Wilcoxon). Where there is imbalance, in covariates, we will use unadjusted and adjusted regression models to ensure that confidence intervals are appropriately quantified.

Minor

Describe the time to event model as "Cox regression". Explain how the proportional hazards assumption will be evaluated and what covariates will you use in the model (or how you will decide what covariates to include) - linked to major point 1

Manuscript revision: We had provided a response to this review comment in our prior submission but had not revised the manuscript to include a clarification on the Cox regression. We have now revised the manuscript to include the following clarification within the methods section:

Cox regression will be used to compare time from enrolment to development of DTG-associated DRMs in different age and viral load strata. The proportional hazards assumption will be evaluated using graphical methods (Schoenfeld residual plots) and a goodness-of-fit test (Schoenfeld Residual Tests). Baseline imbalance between study arms will be assessed, and if present, these variables will be included as covariates in the Cox regression model.

What software will be used?

Manuscript revision: We had provided a response to this review comment in our prior submission but had not revised the manuscript to include a clarification on the statistical software that will be used. We have now revised the manuscript to include the following clarification within the methods section:

Statistical analysis will be performed using R version 4.5.1.

We appreciate your consideration of the revised manuscript.

Sincerely,

Dr. Loice Achieng Ombajo

Principal Investigator,

Ndovu Study

---

## [Decision Letter · Decision Letter 2]

20 Feb 2026

The dolutegravir failure cohort: a multi-country longitudinal cohort with a randomised clinical trial of continued dolutegravir versus switch to darunavir in people with viraemia while on dolutegravir in Sub-Saharan Africa (The Ndovu Study) protocol

PONE-D-25-41434R2

Dear Dr. Loice Achieng Ombajo

We’re pleased to inform you that your manuscript has been judged scientifically suitable for publication and will be formally accepted for publication once it meets all outstanding technical requirements.

Kind regards,

Sarah Nanzigu, Ph.D.,MSc.,MBchB

Academic Editor

PLOS One

Reviewers' comments:

Reviewer's Responses to Questions

**Comments to the Author**

1. Does the manuscript provide a valid rationale for the proposed study, with clearly identified and justified research questions?

Reviewer #1: Yes

2. Is the protocol technically sound and planned in a manner that will lead to a meaningful outcome and allow testing the stated hypotheses?

Reviewer #1: Yes

3. Is the methodology feasible and described in sufficient detail to allow the work to be replicable?

Reviewer #1: Yes

4. Have the authors described where all data underlying the findings will be made available when the study is complete?

Reviewer #1: Yes

5. Is the manuscript presented in an intelligible fashion and written in standard English?

Reviewer #1: Yes

You may also provide optional suggestions and comments to authors that they might find helpful in planning their study.

Reviewer #1: I am satisfied with the authors' responses and the resulting changes to the paper. I have nothing else to add.

**Do you want your identity to be public for this peer review?** For information about this choice, including consent withdrawal, please see our Privacy Policy

Reviewer #1: No

---

## [Editor Report · Acceptance letter]

PONE-D-25-41434R2

PLOS One

Dear Dr. Ombajo,

I'm pleased to inform you that your manuscript has been deemed suitable for publication in PLOS One. Congratulations! Your manuscript is now being handed over to our production team.

Kind regards,

on behalf of

Dr. Sarah Nanzigu

Academic Editor

PLOS One